# Aging-Related Behavioral Patterns in Tibetan Macaques

**DOI:** 10.3390/biology12101325

**Published:** 2023-10-11

**Authors:** Tong Zhang, Shen-Qi Liu, Ying-Na Xia, Bo-Wen Li, Xi Wang, Jin-Hua Li

**Affiliations:** 1School of Resources and Environmental Engineering, Anhui University, Hefei 230601, China; 18297815809@163.com (T.Z.); larry_1206@163.com (S.-Q.L.);; 2International Collaborative Research Center for Huangshan Biodiversity and Tibetan Macaque Behavioral Ecology, Hefei 230601, China; 3School of Life Sciences, Hefei Normal University, Hefei 230601, China

**Keywords:** aging, social behavior, females, partner selection, Tibetan macaques

## Abstract

**Simple Summary:**

The process of aging itself and the behavioral changes caused by aging have been extensively studied and recognized in the field of biology. In this study, we determined that age had no effect on social behavior in male Tibetan macaques (*Macaca thibetana*). Old female macaques were less likely to approach other monkeys. This study provides a new perspective on adjusting social interaction strategies in old non-human primates under nature environments.

**Abstract:**

Aging can induce changes in social behaviors among humans and nonhuman primates (NHPs). Therefore, investigating the aging process in primate species can provide valuable evidence regarding age-related concerns in humans. However, the link between aging and behavioral patterns in nonhuman primates remains poorly comprehended. To address this gap, the present research examined aging-related behaviors exhibited by Tibetan macaques (*Macaca thibetana*) in their natural habitat in Huangshan, China, during the period from October 2020 to June 2021. We collected behavioral data from 25 adult macaques using different data collection methods, including focal animal sampling and ad libitum sampling methods. We found that among adult female macaques, the frequency of being attacked decreased with their age, and that the frequency of approaching other monkeys also decreased as age increased. In males, however, this was not the case. Our findings demonstrate that older female macaques exhibit active conflict avoidance, potentially attributed to a reduction in the frequency of approaching conspecifics and a decreased likelihood of engaging in conflict behaviors. This study provides some important data for investigating aging in NHPs and confirms that *Macaca* can exhibit a preference for social partners under aging-related contexts similar to humans.

## 1. Introduction

In 2006, the World Health Organization (WHO) defined the cut-off age of elderly as over 65 years old for developed countries and 60 years old for developing countries. Globally, the population of individuals aged over 60 years is expected to exceed 20% of the total population by 2050 [1]. In humans, growing older means less social interaction and weaker social networks [2]. This complies with the socioemotional selectivity theory (SST), which posits that as individuals age, they would become more aware of the finite nature of their lives and tend to prioritize relationships with social partners who provide greater value and meaning [3]. A change in social motivation, however, does not appear to be only caused by a constrained perspective on the future [4]. Older female vervets (*Chlorocebus sabaeus*) and chimpanzees (*Pan troglodytes*) have smaller social networks than younger conspecifics, even though they lack awareness of their own mortality [5,6]. It can be seen that aging affects the behavior of social animals, but this process is less.

The behavioral processes of nonhuman primates (NHPs) are comparable to those of humans [7]. Many NHPs, such as Macaca, exhibit age-related behaviors that resemble those observed in humans [8]. For instance, a previous study showed that among wild rhesus macaques (*Macaca mulatta*), older members prefer to focus on specific partners individuals with whom they themselves have stronger bonds [9]. Furthermore, adult female rhesus macaques deliberately restricted the size of their networks as they became older, and concentrated on partners who had previously been associated with fitness benefits, such as relatives and partners who they had been closely and persistently connected to early in life [10]. Clearly, social preferences change with age, not only in humans but also in NHPs.

Most studies have focused on the effects of aging on affiliation behavior in non-human primates, and empirical studies have provided conflicting results. Harrison (2023) [11] asserts that old nonhuman primates concentrate on their social network. For example, a previous study showed that the frequencies of proximity behavior decreases with age in female Toque macaques (*M. sinica*: [12]), but another study showed that among captive stump-tailed macaques (*M. arctoides*), older females increase the distance from other monkeys to minimize the possibility of interaction [13]. In barbary macaques, old individuals were rarely targeted for friendly social interactions, and were seldom approached by others [14]. Regarding social grooming behavior, several studies have found that the frequency of grooming given among animals decreases with increasing age [8,15,16]. For instance, a study conducted on Japanese macaques (*M. fuscata*) observed that as adult female monkeys age, their grooming behavior tends to be directed towards kin, especially low-ranking elderly females [17]. In another investigation exploring the influence of aging on social behavior in Japanese macaques, Kato (1999) [18] discovered that older monkeys exhibit a tendency to distance themselves from their peers, resulting in a decline in the frequency of social interactions and an increase in solitary behavior. However, Pavelka found no correlation between social grooming and aging in older female Japanese macaques [19]. The same set of monkeys from the Nakamichi study were used in this investigation. On the effect of age on the number of social partners in monkeys, many studies have shown a decrease in the number of social partners for tufted capuchin monkeys (*Sapajus* sp.) and barbary macaques (*M. sylvanus*) with increasing age [8,15]. In another study, Pavelka (1990) [20] reported no correlation between the number of social partners and the age of female Japanese macaques. These findings indicated that different species of macaques exhibit different behavioral patterns with increasing age. As such, the effect of age on grooming and proximity behavior needs to be studied in further species of macaques.

It is noteworthy that even the findings of previous studies on the effects of advancing age on aggressive behavior patterns of animals are inconsistent. Previous studies have shown that the aggression levels of chimpanzees decreased with age [21]. Another study on rhesus macaques residing on the island of Cayo Santiago showed that the aggression levels of the monkeys increased with age [22]. A more recent study on Barbary macaques showed that while the aggressiveness of older monkeys was comparable to that of young monkeys, a higher proportion of older monkeys exhibited mild aggression (i.e., threats; [15]). The above examples can illustrate that potential regulators of the primate social aging process include social organization, sex, and dominance status [23]. However, several other studies have found no correlation between age and the aggression levels of monkeys [13,24,25]. Most studies of aggressive behavior in old non-human primates have focused on females, with insufficient studies of males. Specifically, most studies on macaques have only focused on changes in female aging behavior. To fill in the gap, we elucidate the social behavior patterns of older females and males in Tibetan macaque (*Macaca thibetana*). Our research aims to examine the patterns of behavioral interactions, including factors, types, and partner choice, as well as the behavioral functions that primate species exhibit in old age. Specifically, we are interested in understanding how these behaviors contribute to the reduction in aggression and the maintenance of social rank stability.

The Tibetan macaque has a matriarchal society with many females and various males. Females stay in groups, while males tend to migrate in groups at all ages. The females can live up to 30 years in the wild [26]. The research on affiliative and aggressive behaviors of Tibetan macaques at has primarily focused on age-specific behavioral changes observed in immature individuals [27]. In the present study, our aim is to address the behavioral patterns of affiliative and agonistic in old macaques. To do this, we predicted that (1) older monkeys would exhibit less frequency of grooming given and have fewer grooming partners compared to younger adults, (2) older monkeys would have fewer monkeys in proximity, (3) older monkeys would be less often the target of attacks and mostly displayed mild aggression.

## 2. Methods

### 2.1. Study Site and Subjects

The study site was located in the Wild Monkey Valley of Huangshan mountain, Anhui Province, China (30°29′ N, 118°11′ E), at an altitude of 600~1200 m. The study group is habituated to researchers (i.e., from <1 m) and was provisioned daily with 3–4 kg of corn by reserve staff. After feeding, the monkeys leave the provisioned area and continue their natural and undisturbed activities in the forest [26]. The study focused on the YA1 group, which comprised a total of 59 individuals, including 25 adults, 6 sub-adults, and 28 juveniles. Our research team has been conducting a continuous study on this group since 1986, meticulously documenting changes that occur throughout the year, such as births, deaths, and individual dynamics in terms of immigration and emigration. Each member of the group can be identified based on distinctive physical characteristics, and the matrilineal kinship relationships of all animals have been determined. We divided the age groups of male macaques by the color of their coats. There are three categories, including young adults, middle-aged, and old monkeys. No immigration or emigration events occurred in all males during the study period. Female monkeys are usually ready to give birth after age five (mean = 5), and the longest recorded life span for this group of monkeys to date is 30 years old. All 25 adult Tibetan macaques were selected as subjects (October 2020–June 2021). The female monkeys’ age classes were young adults (5–10 years), middle-aged (10–15 years), and old monkeys (>15 years), see Table 1. 

### 2.2. Data Collection and Behavioral Definition

During the study period (October 2020–June 2021, including 100 effective observation days), the group was tracked and observed each day from 08:00 a.m. to 05:00 p.m. The order of focal animals was determined by random drawing, and the sampling time was 15 min. After completing a round of observations in which we recorded data on each focal animal, I reordered the individuals using a random sampling method, and then conducted the next round of observations until 5 p.m. We used a Lenovo D66 recorder (Lenovo China, Beijing, China) to make notes about animal behavior and then entered those notes into Excel (2022) for further study. All behaviors of target animals were observed and recorded within the sampling time using the focal sampling method. A total of 156.25 h (mean = 6.25 h/per individual) of focal animal observations were made. The ad libitum sampling methods were used to record the process of aggressive behavior [28]. In the event of an attack, we recorded the type of aggression, initiator, receiver, response behavior type of the recipient, and whether the third party joined. We categorized the aggression behavior into the following group (Table 2): threat, short lunge, long lunge, chase, and bite. We categorize aggressive behavior types into mild (threat, short lunge, long lunge, and chase) and intense aggression (bite) based on whether there was physical contact [29]. In addition, grooming is a very important behavior in nonhuman primates, as defined in Table 2. We used David’s score (DS) to determine the hierarchy of monkeys in the YA1 group based on the aggression-submission bouts [30].

### 2.3. Data Analysis

We used the individual scores of various social behaviors to analyze the data. The grooming index was used to classify the grooming behavior into grooming given and grooming received. It was calculated as seconds of grooming given or grooming received per hour of observations. The same method was used to quantify proximity index. We calculated two grooming concentration indices as measures of the tendency of the monkeys to initiate/receive their grooming to/from a single preferred grooming partner. The preferred grooming partner’s (the monkey who received or gave the most grooming) grooming to the total grooming given or received was used to generate these indices [8]. To assess the agonistic behaviors, we calculated the number of times aggression was exhibited and aggression was received. The average number of partners within 5 meters during the observation period also was calculated.

For all the data, K-S (Kolmogorov–Smirnov) test was used to analyze its normality; if not, a nonparametric analysis method was used. Given the lack of precise age information for the males, we divided them into distinct age groups and analyzed various behavioral indicators to elucidate the impact of age on social behavior. For male Tibetan macaques, we employed the Kruskal–Wallis test to compare variations in grooming, proximity, and aggression among different age groups. Subsequently, if significant differences were detected, the Dunn test was utilized for multiple comparisons.

The impact of female age on social behavior was investigated using various models. Regarding female individuals, for further analyses of the effect of age on various types of social behavior, we considered “dominance rank” as the potential confounder. For the grooming given and grooming received behaviors, we ran two linear mixed models with age and dominance rank as fixed effects and study group membership as the random effect. The grooming given and groom received indices were entered into the models for requisite analysis.

To understand the influence of age on the frequency of the monkey approach toward other monkeys, we used a generalized linear mixed model (GLMM) with a Poisson response (GLMM1). For GLMM1, we used age and rank as predictive variables, the number of times of approaches as the dependent variable, and focal animal identity as a random factor. Furthermore, to assess the influence of age on the frequency of a monkey being the object of approach, we used another GLMM with a Poisson response (GLMM2). For GLMM2, we used age and rank as predictive variables, the time of being approached as the dependent variable, and focal individual identity as a random factor. In addition, we used a third GLMM with a Poisson response (GLMM3) to explore the influence of age on the average number of monkeys within 5 m of the focal monkey. For GLMM3, we used age and rank as predictors, the average number of monkeys within 5 m of the focal monkey as the dependent variable, and focal individual identity as a random factor. To explore the influence of age on the frequency of monkeys receiving aggression, we used a GLMM with a binomial response (GLMM4). For GLMM4, we used age and dominance as predictors, the aggression received as the dependent variable, and treated individual identity as a random factor. To determine the effect of age on the proportion of mild aggression in the total aggression involving older monkeys, we used another GLMM with a binomial response (GLMM5). In GLMM5, we included age and dominance rank as covariates, treating focal animals as varying intercepts. To examine the impact of age on grooming concentration indices, we employed a linear mixed model. The model incorporated age, dominance rank, and the interaction between age and dominance rank as fixed effects, while accounting for social group membership as the random effect. To determine whether grooming behavior is associated with the dominance rank, we used partial correlation analysis to identify a potential relationship between a monkey’s grooming and the rank of the grooming recipient, while adjusting for the sex of the grooming recipient. To determine whether older monkeys groomed their relatives first (coefficient of kinship = 0.5: mother; 0.25: grandmother, siblings; 0.125: aunts, uncles, nephews, nieces; and 0.063: cousins; [31]), partial correlation analysis was used to explore the relationship between a monkey’s grooming and its kinship coefficient with the recipient, while controlling for the rank of the recipient. We also used the partial correlation analysis to assess the relationship between a monkey’s grooming and the grooming received by other monkeys to explore the dyadic reciprocity of the target group while adjusting for the kinship of the recipients. Pearson’s correlation analysis was used to explore the relationship between proximity and total number of aggressive acts (*N* = 361).

Before substituting all dependent variables into the model analysis, the corresponding methods were used to check whether they conform to the data distribution type required by the model. We ran all models in R package lme4 (R Core Team 2020; version 4.0.3) using the function “glmer” (GLMM applied to binomial-and Poisson-dependent variables; [32]). Kruskal–Wallis test, K-S test, Pearson’s and Spearman’s correlation analyses, and partial correlation analyses were performed in SPSS (22.0). The level of significance was set at α < 0.05. ORIGINPRO (9.6.5.169) was used to create all figures.

### 2.4. Ethics Statement

This study complies with the regulations of the Chinese Wildlife Conservation Association regarding the ethical treatment of research subjects and the law of the People’s Republic of China regarding the protection of wildlife. The study was conducted purely through observational methods, ensuring that data collection had no impact on the welfare of the monkeys involved. Huangshan Monkey Management Center and the Huangshan Garden Forest Bureau permitted us to conduct research at the field site.

## 3. Results

### 3.1. Influence of Age on Affiliative Behavior

Males engaged in various affiliative social behaviors. However, the results of K-W test analysis showed no significant differences in social behavior among males in the three age groups (young/middle/old; Table 3).

As for female monkeys, the dominance rank did not affect the frequency of grooming given (linear mixed model: *t* = −0.776, *N* = 13, *p* = 0.453). Furthermore, age and rank also did not affect the frequency of grooming received (linear mixed model: age: *t* = 0.127, *N* = 13, *p* = 0.901; rank: *t* = 0.289, *N* = 13, *p* = 0.778). There was no correlation between the preference for high-rank monkeys (partial correlation: *r* = 0.152, *N* = 15, *p* = 0.605), and relatives (partial correlation: *r* = 0.352, *N* = 15, *p* = 0.251), and reciprocity among grooming recipients (Partial correlation: *r* = 0.045, *N* = 15, *p* = 0.879). We found that age did not affect grooming concentration indices (linear mixed model: grooming given concentration index: SD: 0.011, *Z* = 1.615, *p* = 0.082; grooming received concentration index: SD: 0.009, *Z* = 1.102, *p* = 0.074).

We observed a significant decrease in the frequency of approach toward other partners as the monkeys aged (GLMM1: est. = −0.071, *SE* = 0.029, *p* = 0.018, Table 4, Figure 1). Rank did not affect the frequency of approach to other monkeys (GLMM1: est. = −0.078, *SE* = 0.043, *p* = 0.070, Table 4). Moreover, age and rank did not affect the rates of being the object of approach (Table 4, GLMM2). We did not observe any effect of age on the average number of animals within 5 meters per focal animal (GLMM3: est. = −0.048, *SE* = 0.061, *p* = 0.429, Table 4). In addition, for the old female monkey, we observed a significant correlation between the total number of aggressive acts and proximity (Pearson: *r* = 0.346, *N* = 115, *p* < 0.001, Figure 2).

### 3.2. Influence of Age on Aggressive Behavior

Additionally, we found no noticeable disparity in aggressive behavior among different age groups of males (Table 1). With respect to females’ aggressive behaviors, GLMM analysis revealed that age affected the frequency of the aggression received (GLMM4: *SE* = 0.128, *Z* = –2.696, *p* < 0.001, Table 4, Figure 3), that is, older individuals would be attacked less often. We observed that rank also affects the frequency of aggression received (GLMM4: est. = 0.277, *SE* = 0.129, *p* = 0.033, Table 4), with higher-ranking individuals experiencing fewer attacks. However, we did not observe any impact of age on the proportion of mild aggression in all of aggressive behavior exhibited by the monkeys (GLMM5: est. = 0.011, *SE* = 0.001, *p* = 0.619, Table 4). In addition, in older monkeys, we found a significant correlation between the total number of aggressive acts and the proximity index (Spearman correlation: *r* = 0.349, *N* = 151, *p* < 0.001).

## 4. Discussion

This study examined the effects of aging on social behavior in Tibetan macaques. No evidence was found in this study to support prediction 1 (older monkeys would exhibit less frequent grooming given and have fewer grooming partners compared to younger adults). Old female macaques were less likely to approach other monkeys, supporting prediction 2 (older monkeys would have fewer monkeys in proximity). Social behavior did not change with age in male macaques. However, the older the monkeys, the less likely they were to be the target of aggression, partially supporting prediction 3 (older monkeys would be less commonly the target of attacks and mostly display mild aggression). Female Tibetan macaques have an aging process similar to that of humans. In the study, female macaques failed to present obvious partner choice, but approached other monkeys less. These findings suggest that Tibetan macaques can exhibit changes in social behavior associated with aging.

We found that the older the female Tibetan macaques were, the less often they approached other monkeys. In this despotic species, high-ranking monkeys exhibit limited tolerance for other monkeys, and conflict interactions may occur when they approach other monkeys [29]. Older female Tibetan macaques rarely approach other individuals, possibly to reduce the risk of being attacked. Furthermore, we did not observe any significant reduction in the frequency of grooming behavior with increasing age. Similarly, a previous study reported no negative impact of the age of female Japanese monkeys on their social behavior [19]. However, a study on a similar group of Japanese monkeys yielded contrasting results. Nakamichi (2003) [17] discovered that as Japanese monkeys aged, their grooming behavior tended to be directed more toward related individuals and less toward unrelated ones. Likewise, other studies [13,18] have observed social withdrawal in older Japanese monkeys. However, certain studies have reported a decline in proactive grooming behavior among older monkeys [8,15]. Our results suggest that old female Tibetan macaques actively choose to avoid conflict.

In our study group, we observed a negative correlation between the age of the monkeys and the frequency of them being the target of aggression. This result was in agreement with the findings of the previous studies. Studies have shown that as rhesus macaques became older, they were attacked less often [21]. This may be explained by the fact that older monkeys approach other individuals less actively. Furthermore, our study observed no effect of age on the proportion of mild aggression in the total aggressive behavior the monkeys were engaged in. The lack of correlation between mild aggressive behavior and age may be attributed to the high prevalence of mild aggression at the group level. In addition, we found that in female Tibetan macaques, the total number of aggressive acts positively correlated with the proximity index.

In non-human primates, proximity increases the chance of friendly contact between individuals, it also increases the probability of conflict between individuals [33]. Our results showed that the more time spent in proximity of female macaques, the more aggressive behaviors occurred between individuals, especially those who are related. It is well known that in non-human primates, related individuals spend a lot of time engaging in affiliative social behaviors such as grooming, but conflict is also inevitable. This agrees with what we discovered. The Tibetan macaque is a species with a despotic dominance style. This suggests that in the Tibetan macaque population, in addition to low conciliatory tendencies, the macaques consistently displayed highly asymmetric patterns of aggression and little counter aggression [29], and the process of approaching another animal may easily lead to conflict. Perhaps this could explain why individuals with long proximity time with each other attack more regularly.

In our study, no significant differences were found in social behavior between older male Tibetan macaques and other age groups. Due to the unknown age of male individuals, it is unlikely that the effects of age on social behavior can be delved into. Also, comparison studies based on different rankings have not been conducted due to only having three old male individuals. Tibetan macaques exhibit a despotic social structure, where females tend to remain in the group while males have the potential to emigrate during adolescence [29]. Consequently, the age range can only be estimated based on facial features and hair color, and accurate age determination is not possible. It is plausible that the absence of significant changes in social behavior among older males may be attributed to the fact that they are not considered to be in the advanced age category.

In the current study, we did not observe a significant decrease in the average number of monkeys within 5 m of female monkeys with increasing age. Our findings do not corroborate those of previous studies. In a previous study on Barbary macaques in Rocamadour, the average number of monkeys within 5 m of the focal animal decreased with age [15]. In the study, age had no effect on grooming given in female macaques. It can be explained by the studies about older high-ranking female monkeys being as socially attractive as younger high-ranking females [17]. In the study, there were three 18-year-old monkeys and one 29-year-old monkey in the group, along with two high-ranking animals. We hypothesize that we did not observe an adverse effect of age on active grooming and the number of partners within 5 m of the monkeys because the monkeys in our group did not reach the late stages of old age, and their social style was despotic dominance style.

Among humans, older individuals use several strategies to actively avoid negative social interactions [34]. In humans, physical decline with age is synchronized; animals undergo behavioral changes as they age to conserve energy [35]. In our study, older female macaques did not exhibit a preference for any particular partners, but they also decreased the frequency of approaching other people, suggesting that they might be avoiding unfavorable social contacts.

Taken together, our findings suggest that older female Tibetan macaques exhibit a reduced inclination for active engagement with other monkeys. This behavior contributed to a lower frequency of conflict compared to younger adult monkeys. Furthermore, the monkeys that engaged in the most conflict with older monkeys were those in close proximity to the older females. Overall, our results support the socio-emotional selectivity theory to some extent. Our findings shed light on data on behavioral patterns in age-related contexts in nonhuman primates. This may help to better understand the effects of aging on humans. Future research may focus on how older monkeys trade off partner choice during their social interactions.

## 5. Conclusions

In conclusion, aging has an impact on the social behavior of Tibetan macaques. Our results show that older female macaques are less likely to approach other individuals, and they are less likely to be attacked. As a result, older female macaques flexibly adjust their behavioral strategies to avoid conflict. However, aging had no effect on the social behavior of male macaques. This partly explains why non-human primates can also produce aging changes in social behavioral choices without a sense of the future. This study adds to the growing body of evidence on the effects of aging on social behavior in nonhuman primates. However, this study was a cross-sectional one, and future research should focus more on how age-induced behavioral changes occur.

## Figures and Tables

**Figure 1 biology-12-01325-f001:**
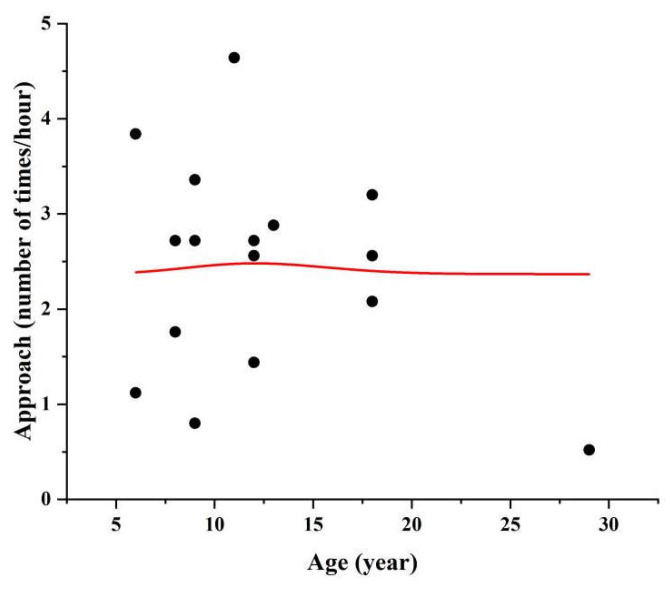
Effect of age (years) on the number of times of approach in female macaques.

**Figure 2 biology-12-01325-f002:**
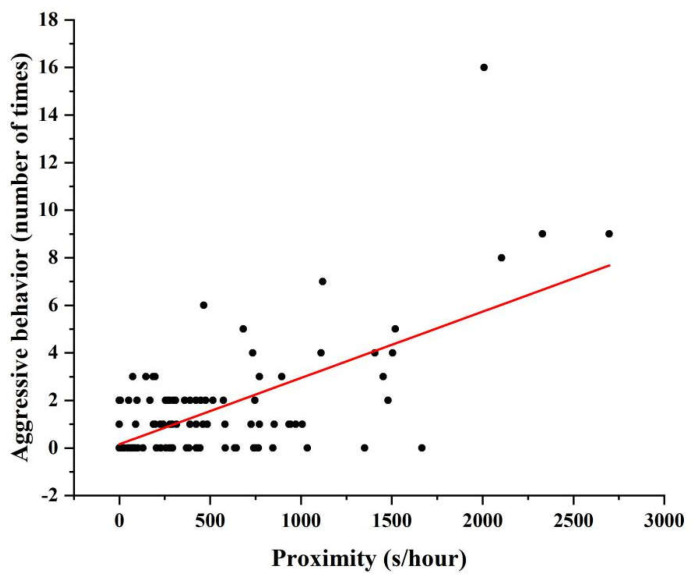
Correlation between proximity frequency and number of attacks in female macaques.

**Figure 3 biology-12-01325-f003:**
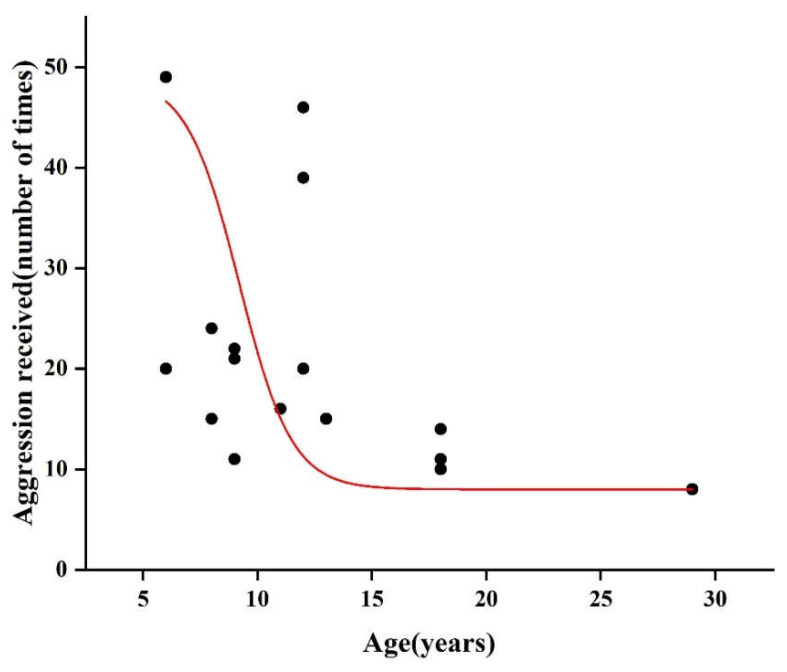
Effect of age (years) on the aggression received (the number of times) in female macaques.

**Table 1 biology-12-01325-t001:** Composition of YA1 group of Tibetan macaques.

Male ID	Social Rank (DS)	Age Group	Mother	Immigrate/Birth Time	Female ID	Social Rank (DS)	Age	Mother
YXK	1 (27.00)	Middle	YH	14 February 2013	YXX	1 (81.00)	11	YH
TRG	2 (19.88)	Middle	#	24 April 2010	YH	2 (67.67)	18	YM
ZB	3 (13.00)	Old	#	25 August 2008	YCY	3 (56.67)	12	YM
DS	4 (4.71)	Old	#	10 August 2013	YXY	4 (40.75)	6	YH
TQ	5 (−3.33)	Middle	#	27 November 2018	YM	5 (35.37)	29	#
WM	6 (−3.67)	Middle	#	20 November 2018	YCH	6 (30.00)	9	#
TQS	7 (−10.83)	Young	TXH	4 May 2015	TXH	7 (9.90)	12	TH
JM	8 (−19.88)	Young	#	3 September 2020	YCL	8 (8.17)	9	YM
SJT	9 (−26.88)	Old	#	18 September 2020	TH	9 (−10.00)	18	TG *
					TXX	10 (−23.77)	13	TH
					TQL	11 (−36.00)	8	TXX
					HH	12 (−44.53)	18	H *
					THY	13 (−45.27)	12	TR *
					HXY	14 (−45.46)	6	HH
					THX	15 (−63.58)	9	TH
					HXW	16 (−80.57)	8	HH

**#**: Mother is unknown. *: Mother is dead or not in the YA1 group. DS: David’s score.

**Table 2 biology-12-01325-t002:** Definition of social behaviors observed during the study period.

Behavior	Definition
**Affiliative**	
Grooming	An individual uses his/her fingers and palms to groom the fur of another individual. The groomer may pick out small objects from the recipient’s fur and eat them.
Proximity	Two or more individuals keep a sitting or lying posture within a certain distance; the distance in this study was 2 m.
Approach	Focal subject came from beyond to ≤1 m of another individual, or vice versa.
**Aggressive**	
Threat	An individual directs an open mouth threat gesture or any of its components, e.g., stare, raised eyebrows, lowered jaw, ground slap, to another individual.
Short lunge	An individual directs a lunge <2 body lengths to another individual.
Long lunge	An individual directs a lunge >2 body lengths to another individual but does not go into a full chase.
Chase	An individual runs rapidly after another individual.
Bite	An individual grabs and bites hard, either releasing the victim quickly or hanging on for several seconds.

**Table 3 biology-12-01325-t003:** Differences in male social behavior among different age groups.

	Grooming Given	Grooming Received	Approach Given	Approach Received	Aggression Given	Aggression Given	Grooming Given Concentration Index	Grooming Received Concentration Index
X^2^	1.238	2.250	4.950	2.250	2.285	0.375	0.225	1.238
*df*	2	2	2	2	2	2	2	2
*p*	0.539	0.325	0.084	0.425	0.325	0.829	0.894	0.539

**Table 4 biology-12-01325-t004:** GLMMs for explaining the impact of age on the frequency of approach (GLMM1), the frequency of being the object of approach (GLMM2), the average number of monkeys within 5 meters (GLMM3), the number of aggressive acts received (GLMM4), and on the proportion of mild aggression in all aggression behaviors female monkeys were engaged in (GLMM5).

Model	Predictors	Estimate	*SE*	*Z*	*p*
GLMM1: Frequency of approach					
	Intercept	3.919	0.375	10.446	<0.001
	Age	−0.071	−0.029	2.360	0.018
	Rank	−0.078	0.043	−1.812	0.070
GLMM2: Frequency of being the object of approach					
	Intercept	2.717	0.371	7.319	<0.001
	Age	−0.005	0.026	−0.187	0.852
	Rank	−0.012	0.041	−0.283	0.777
GLMM3: The average number of monkeys					
	Intercept	1.982	0.768	2.582	0.009
	Age	−0.048	0.061	−0.791	0.429
	Rank	−0.046	0.089	−0.518	0.605
GLMM4: The number of aggressive acts received					
	Intercept	4.985	1.367	3.647	<0.001
	Age	−0.346	0.128	−2.696	0.007
	Rank	−0.277	0.129	−2.135	0.033
GLMM5: The proportion of mild aggression					
	Intercept	0.481	0.117	4.098	0.002
	Age	0.011	0.001	0.597	0.619
	Rank	0.104	1.001	0.104	0.917

## Data Availability

Data are available on request.

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
