# Peer review of "Aging-Related Behavioral Patterns in Tibetan Macaques"

_biology, 2023, doi:10.3390/biology12101325_

Round 1
Reviewer 1 Report
This manuscript addresses an interesting topic, specifically whether social affiliative behaviors change with age in Tibetan macaques. In general, the introduction is a bit weak – for example, the implications of the socio-emotional selectivity theory (SST) are not well explained. The SST is also poorly addressed in the discussion, being mentioned at the end without directly addressing the implications of the results presented here for this theory. These problems may be because this theory is not a very good fit for the data presented here, which provide a straightforward example of both changes and the absence of changes in social behavior with age.
The discussion is also poorly integrated. If I understand the statistics correctly, the methods here are somewhat problematic because behaviors appear to have been analyzed individually rather than in one overall analysis (I realize that some of the distributions differed and some but not all responses were binomial, thus requiring separate analyses); this suggests that the unadjusted p values would be inflated.
It is not clear how this study ‘provides a new prospective on adjusting social interaction strategies in old non-human primates under competitive environments.’ There are no data here demonstrating that there is a competitive environment, for example.
Nonetheless, the information presented here is of interest to a wide audience, and with appropriate revisions this paper can provide an interesting addition to the literature.
Specific comments:
Ll 11-12 Should read ‘that age had no effect on social behavior in male Tibetan macaques (Macaca thibetana).’
Ll 25-26. ‘had the behavioral patterns of selection of social partners’ does not make sense in English
Ll 47-48. Do you mean that the older macaques prefer to focus on individuals that have generally stronger social bonds or individuals with whom they themselves have stronger bonds?
L. 55. What does ‘the proximity behavior decrease’ mean? Also, correct grammar.
L. 99. ‘Try to’ is not appropriate here. And it sounds as though you are making a comparison between younger and older macaques.
L 123. In Table 1, what does ‘DS’ stand for?
L. 128-129. Please clarify. I think you mean this: The order in which we observed focal individuals was determined at the beginning of each day by a random drawing. Each focal observation lasted 15 min, and after completing a round of observations in which we recorded data on each focal animal, we began again at the top of the order. We typically performed XX observations on each individual daily.
L. 134. How did you do both ad libitum and all-occasion sampling? Did you not record aggression whenever you saw it?
L. 137. What definition? Just write something similar to the following: ‘We categorized aggression into the following groups:…. (reference).
L. 138. Threatening. Or just use ‘threat.’
Table 2. Be sure that each definition is aligned with the behavior it describes. i.e. they should start on the same line.
L. 144. Does ‘s/hour’ here mean seconds per hour? If so this is not a frequency, which should be the number of times something occurred.
The term ‘active grooming’ is confusing because the grooming is active whether the focal animal is grooming someone or someone is grooming the focal animal.
L. 153. What is ‘partner’ here? Does that refer only to grooming?
Ll. 155-6. Rewrite for clarity.
L. 169. The ‘monkey acting approach?’
L. 172. Do you mean the time at which approaches occurred or their duration?
L. 218-220. No significant differences among what? This section is confusing because It is headed ‘influence of age on affiliative behavior,’ but the second sentence under the header says ‘Since it was unknown what age the male monkeys were, the model could not be used to examine how age affect their social behaviors.’ What model?? Then the table just under this (Table 3) is titled ‘Differences in male social behavior among different age groups? To make this more clear, include the age groups you used here as a reminder and instead of writing about age, use ‘age group.’ E.g., you could start by stating that in males, age group had no effect on the frequency of the social behaviors we tested.
The results for females just below are also written unclearly. If, for example, rank did not affect the frequency of grooming females received (l. 225), then why did you test whether there was a correlation between highly ranked monkeys and relatives? Perhaps the problem here is just that this is not clearly explained?
L 235, 244, etc. What is an ‘acting approach?’
L. 246. Aggression can’t be counted – language problem here.
L. 250. This is about males. However, Fig. 2 is a graph showing the effects of age on aggressive actions received by females. Fig. 2 is also identical to Fig 1.
L. 267. Why are the numbers ‘1,’ ‘2,’ and ‘3’ in this paragraph? They aren’t in order, so it can’t be a numbered list.
L. 269. Restate hypotheses in Discussion rather than just giving their numbers. And why start with Hypothesis 2?
Reword: ‘We found no changing in aging social behavior….’ Probably you mean ‘Social behavior did not change with age in male macaques.’ The numbers of approaches to others did not significantly differ with male age, so if you want to bring that behavior up here (as you did) then you should point that out rather than implying that it is a significant difference.
L. 290. What are ‘conflicting events?’
L. 302. What is a ‘strict dominance style??’ Please be more specific here.
L. 319. Not sure what this means: We observed no significant difference in the overall acting grooming level of the 319 study group.
L. 321-325. Here briefly explain what ‘old’ is for this species. Also, what does ‘their social style was strict’ mean?
L. 327-330. More information is needed here if you are going to compare these monkeys to humans, and in any case it would be more informative to compare more broadly across a wider range of non-human primates.
The usage of English is in general very good (as is the vocabulary), but in some places in the manuscript there are difficulties with language that affect the readability and meaning of the writing. I would recommend this paper be edited by someone who is a native English speaker or who has a very high level of skill in that language.
Reviewer 2 Report
I commend the authors for their efforts in conducting this research, which addresses the important and timely scientific question of social aging in nonhuman primates. However, after a careful and thorough review of the manuscript, I cannot recommend it for publication in its current form.
While the manuscript presents several noteworthy findings and contributions, it also contains substantial issues that must be addressed before it can be considered for publication. In its present state, the manuscript lacks the necessary literature review, clarity, and organization. I believe that with significant revisions and improvements, this work would make a valuable contribution to the scientific literature. However, I request that the authors undertake major revisions to address the following concerns:
Literature Review: In its current form, the manuscript's review of relevant literature is limited and lacks depth. The authors should expand their literature review to provide a more comprehensive understanding of the context in which their research is situated. For example, the introduction is missing several recent papers, reviews, and commentaries outlining the expansion of research in social aging in nonhuman primates, manuscripts missing include Machanda et al. 2020, Fischer 2022, Negrey et al. 2023, Harrison 2023, and Newman et al. 2023…etc. The authors should rely heavily on data from the Cayo Santiago population of rhesus macaques, which provides an important comparative opportunity to study aging in a provisioned macaque population. Furthermore, the authors briefly mention social aging in other animal species, but mention a single paper on aging beagles. This feels out of place. The authors should either 1) expand on the phenomenon of social aging across mammals (to better integrate this citation), or 2) focus on the literature exploring social aging in nonhuman primate species.
Clarity of Presentation: The tables and figures need to be revised substantially. Additionally, there are repeated instances throughout the Methods where the authors refer to the incorrect table. This must be corrected. The authors might consider the following suggestions re. the figures & tables:
Table 1. The text provides a vague description of how the study subjects were partitioned into the age categories (Line 120-122). What this based on physical characteristics? Time that the individual was a member of the social group? Additionally, as written, the text suggests that individuals of both sexes were placed in these age categories. The authors do mention the lack of exact aging data for males, but this comes much later in the Methods section. It should be moved up to improve the flow.
Table 2 is incomplete and only includes a fraction of the social behaviors examined during this study. It would be helpful to the readers to separate these behaviors into those that were continuously observed versus those that were collected using ad libitum/point sampling. Accordingly, the authors mention mild aggression on line 184, but they have not defined this term.
Figure 1 & 2 are the same. Figure 1 should be showing data on approaches, not aggression.
Table 4 would be greatly improved if the authors added a column explaining the outcome variables for each model. It is currently listed in the text accompanying the Table.
Why is there no figure illustrating the significant correlation between aggression and proximity? This seems to be a main point of the paper.
Minor point – Lines 172 & 175, I believe the authors mean to say number/counts instead of “time” regarding the counts of approaches to a given subject.
There are some points of confusion in the explanation of the statistical analyses that require attention. For example, the authors state that they determined the interaction between age and dominance rank in their models for grooming (line: 166). However, there is no other mention of the results of this interaction in the results section. Additionally, it is unclear when the interaction term was not included in subsequent modelling.
Organization: The overall organization and flow of the manuscript require substantial improvement. The introduction can be greatly improved by clearly outlining what is known in female versus male nonhuman primates. The authors also can improve their descriptions of the sex differences in social aging in the discussion.
There are several instances where the authors would benefit from editorial oversight for English grammar/phrasing.
Round 2
Reviewer 2 Report
The manuscript is greatly improved from its first version. There are minor suggestions that would help to improve the manuscript further.
Line 121 - please report the average age at first birth for members of this population.
The description of how you approximated male ages feels out of place (line 176 - 178). Given the differences in the statistical approaches between males and females, the methods section would read more clearly if you described the analyses of males and females separately.
Figure 1. The fitted line doesn't seem to be correct. You should include the fitted Poisson trend line to better illustrate the relationship between age and approach frequency.
Line 175 - I do not understand what the numbers in the brackets indicated. Please remove them or explain to what these numbers refer.
I would consider adding another figure to the manuscript illustrating the significant correlation between the number of aggressive encounters and proximity to conspecifics. Also, line 257, you should report p<0.001 instead of p=0.000.
Lastly, there is no mention of the interrelationships between individuals in the discussion. Based on the information in Table 1, many of the study subjects are closely related to each other. Could your results be explained by familial interactions/coalitions?
There are several cases throughout the manuscript where the quality of the english can be improved. Some examples can be found on the following lines:45, 26, 103, 295.
